# Metabolic potentiation of antibiotic killing by L-arginine in drug-resistant *Edwardsiella tarda*

Bei-bei Yan,[1,2] Na Li,[3] Yang Zhou,[4] Li-li Kang,[1,2] Xue-sa Dong,[3] Xiao Xu,[3] Li An,[3] Qing-lei Meng,[3] Xi-rong Wang,[3] Ling Yang,[3] Xiao-ying Li,[1,2] Chao Wang[3]

**ABSTRACT** The role of metabolic state reprogramming in modulating antibiotic susceptibility has attracted growing interest as a promising strategy to combat antimicrobial resistance. Our study revealed that L-arginine potentiates chloramphenicol's bactericidal activity by at least two orders of magnitude against multidrug-resistant *Edwardsiella tarda* via the coordinated modulation of three interconnected metabolic pathways: the tricarboxylic acid cycle disruption, redox homeostasis alteration, and phenylalanine metabolic suppression. Mechanistically, L-arginine-mediated tricarboxylic acid cycle inhibition diminished NADH production and compromised proton motive force, thereby depleting cellular energy supply and impairing drug efflux capacity. Concurrently, L-arginine disturbed the bacterial redox balance, which normally provides antibiotic resistance, by both lowering total antioxidant capacity and raising reactive oxygen species production. Furthermore, L-arginine suppressed phenylalanine metabolism, whereas trans-cinnamate restored antioxidant defenses and proton motive force, diminishing antibiotic resistance. These findings expanded the understanding of metabolic modulation's role in combating antibiotic resistance and offered theoretical support for the development of new antimicrobial strategies.

**IMPORTANCE** The global crisis of antimicrobial resistance demands innovative strategies to revitalize existing antibiotics. Our work addresses this urgent need by demonstrating that L-arginine acts as a powerful potentiator of chloramphenicol, enhancing its bactericidal efficacy by over 100-fold against multidrug-resistant *Edwardsiella tarda*. More significantly, we elucidate a novel, dual-pathway mechanism: arginine concurrently disrupts the TCA cycle and phenylalanine metabolism, which collectively alter the cellular redox state and compromise the proton motive force. This study is the first to uncover this sophisticated metabolic interplay, providing not only a promising adjuvant strategy but also a new conceptual framework for combating resistant bacterial infections by targeting core metabolism. Our findings, therefore, hold substantial potential for both basic science and translational antimicrobial development.

**KEYWORDS** *Edwardsiella tarda*, L-arginine, trans-cinnamate, the tricarboxylic acid cycle, proton motive force

Address correspondence to Chao Wang, wangchao850303@163.com, or Xiao-ying Li, lxy_jn@email.sdu.edu.cn.

The authors declare no conflict of interest.

*E*dwardsella tarda is an intracellular Gram-negative bacterium with a broad host range, capable of infecting various organisms, including humans, birds, and fish. Such infections result in substantial economic losses globally (1, 2). Antibiotics remain the cornerstone of treatment for bacterial infections. However, the overuse and misuse of antibiotics have led to a worldwide surge in antimicrobial resistance (3). Multidrug-resistant bacteria present a significant challenge, severely compromising the efficacy of infection prevention and treatment. In 2019, bacterial antimicrobial resistance was associated with approximately 5 million deaths (4), a number projected to

rise dramatically by 2050 without effective interventions (5). Bacterial antimicrobial resistance, recognized as a critical global health threat, requires a comprehensive understanding of the molecular mechanisms underlying resistance to develop effective strategies against antibiotic-resistant pathogens.

In addition to traditional antibiotic resistance mechanisms, such as reduced membrane permeability, increased efflux pump activity, enzymatic degradation, and drug target modifications (6), metabolic modulation has emerged as an additional factor in bacterial resistance (7, 8). Accumulating evidence indicates that metabolic perturbations play a critical role in antibiotic-induced bacterial death (9, 10). Notably, exogenous metabolite supplementation has emerged as a promising approach to resensitize resistant pathogens through targeted metabolic reprogramming. For instance, in multidrug-resistant *Escherichia coli*, exogenous glutamine enhances membrane permeability and ampicillin uptake, ultimately exceeding the capacity of efflux systems and reducing bacterial survival (8). Termed metabolic reprogramming, this strategy aims to restore the efficacy of existing antibiotics and develop next-generation therapeutics by enhancing their bactericidal effects against resistant pathogens (11–14).

The tricarboxylic acid (TCA) cycle plays a pivotal role in metabolic modulation, serving as a central hub that integrates carbohydrate, amino acid, and lipid metabolism while generating NADH to fuel ATP synthesis. On one hand, the upregulation of the TCA cycle enhances the proton motive force (PMF), facilitating the intracellular uptake of aminoglycosides and thereby improving antibiotic bactericidal efficiency (8, 15–17). On the other hand, the TCA cycle impacts redox balance through reactive oxygen species (ROS) generation from its key product, NADH, during electron transport (18). The effects of different metabolites on the bactericidal activity and mechanisms of antibiotics vary considerably. For example, fumarate enhances the bactericidal activity of aminoglycosides against antibiotic-resistant *P. aeruginosa*, while glycolytic metabolites facilitate the eradication of persistent *E. coli* and *Staphylococcus aureus* (16, 19). However, certain metabolites within the TCA cycle can also promote antibiotic resistance, such as conferring resistance to chloramphenicol (CAP) in *E. tarda* (20, 21).

Beyond their primary protein-targeting mechanisms, antibiotics like CAP, tetracyclines, and quinolones can kill bacteria by stimulating the production of ROS (22–24). The bactericidal efficacy of cefoperazone-sulbactam is further enhanced through metabolic reprogramming of the P cycle, which increases NADH availability for the electron transport chain and consequently stimulates ROS production (25). Conversely, sublethal levels of ROS may promote antibiotic resistance through mechanisms such as induction of multidrug efflux pumps or enhanced mutagenesis resulting from redirected glucose flux into the pentose phosphate pathway (12, 26). This dual role makes it essential to clarify the context-dependent functions of ROS in antibiotic efficacy. To counteract ROS-mediated damage, bacteria rely on the antioxidant glutathione (GSH) to maintain cytoplasmic redox homeostasis. Exogenous pyruvate enhances flux through glycine, serine, and threonine metabolism and cysteine and methionine metabolism by activating cystathionine γ-lyase and glutamic-oxaloacetic transaminase. This metabolic reprogramming disrupts glutathione (GSH) homeostasis, resulting in ROS accumulation that ultimately potentiates gentamicin-mediated killing. Conversely, in antibiotic-resistant strains, glycine downregulation stimulates glutathione reductase activity, facilitating GSH regeneration and subsequent ROS clearance (27, 28). These findings underscore the complexity of the TCA cycle and redox state in modulating bacterial survival and antibiotic susceptibility.

As an essential amino acid, L-arginine (Arg) plays a crucial role in numerous physiological and metabolic processes, including its influence on antibiotic resistance. Arg has been shown to enhance gentamicin-mediated killing of *P. aeruginosa* and *S. aureus* in a pH-independent manner; however, this effect is mitigated by carbonylcyanide-3-chlorophenylhydrazone (CCCP), a proton ionophore that disrupts the PMF. Conversely, Arg's ability to potentiate gentamicin killing of *E. coli* is pH-dependent (29). Additionally, Arg serves as a precursor for nitric oxide (NO) synthesis, which not only modulates ROS but

also diminishes the alanine-enhanced bactericidal effect of gentamicin against *Vibrio alginolyticus* (30). Despite these insights, the mechanisms underlying Arg's impact on the bactericidal efficacy of antibiotics remain incompletely understood, particularly in the context of CAP and *E. tarda*.

In this study, we elucidated how the Arg-related pathway enhances the bactericidal activity of CAP against multidrug-resistant *E. tarda*. We further demonstrated that this effect is intricately linked to the TCA cycle, phenylalanine metabolism, and the regulation of ROS. These findings provide new insights into the bactericidal mechanisms of antibiotics and the regulatory pathways influencing ROS, offering a broader understanding of their role in combating antibiotic resistance.

## MATERIALS AND METHODS

### Bacterial strains and culture conditions

*E. tarda* ATCC 15947 was obtained from the China Center of Industrial Culture Collection. The wild-type strains of BW25113 and isogenic deletion mutants of isocitrate dehydrogenase (Δ*icd*) and α-ketoglutarate dehydrogenase (Δ*sucA*) were obtained from the KEIO collection (31). The strain was cultured in tryptic soy broth (TSB) medium or M9 minimal medium (supplemented with 1 mM MgSO$_4$ and 0.1 mM CaCl$_2$) at 37°C. Exogenous metabolites and antibiotics used in this study, including proline (Pro), ornithine (Orn), citrulline (Cit), gamma-aminobutyric acid (GABA), L-arginine (Arg), D-arginine (D-Arg), putrescine (Put), spermidine (Spd), spermine (Spm), trans-cinnamate (Cin), kanamycin (KAN), chloramphenicol (CAP), ceftazidime (CAZ), and ciprofloxacin (CIP), were purchased from Sigma-Aldrich Corporation.

### Antibiotic bactericidal assays

Bacterial cells were grown overnight in TSB medium at 37°C and harvested by centrifugation at 5,000 × *g* for 5 min. The collected cells were washed three times with sterile saline and resuspended in M9 medium to a final concentration of 10$^6$ CFU/mL. Exogenous metabolites and/or CAP were added to the medium, and the cultures were incubated at 37°C for 5 h. Finally, 10 µL of the culture was serially diluted and plated onto TSA plates. The plates were incubated at 37°C for 18 h, after which bacterial colonies were counted, and the CFU/mL was determined.

### Metabolomic profiling

The sample preparation and metabolomics data acquisition were conducted following the protocol described (21). In brief, bacterial cells were incubated in M9 medium with or without exogenous Arg at 37°C for 5 h. The metabolic reactions were terminated by adding cold methanol, and the cells were washed twice with cold sterile saline. The harvested cells were rapidly frozen in liquid nitrogen and stored at −80°C until analysis. Cell pellets were resuspended in 0.1 mL of prechilled methanol (−20°C) and incubated for 60 min. After extraction, samples were centrifuged at 14,000 × *g* for 15 min at 4°C. The resulting supernatants were concentrated to dryness using a centrifugal evaporator. Dried metabolites were reconstituted in 80% (vol/vol) methanol for subsequent LC-MS/MS analysis. Chromatographic separation was performed on a Vanquish UHPLC system (Thermo Fisher Scientific) coupled to a Q Exactive HF-X hybrid quadrupole-Orbitrap mass spectrometer (Thermo Fisher Scientific). Instrumental parameters were set according to previously established methods (21).

### Metabolomic data analysis

Raw mass spectrometry data were processed using Compound Discover software (version 3.0, Thermo Fisher Scientific). Following background ion removal and normalization using quality control (QC) samples, the identified and quantified peak data were

subjected to subsequent statistical analysis. Differential metabolites were identified based on a threshold of $P$-value < 0.05 and |log2 (fold change)| > 1.5. Metabolic pathway annotation and analysis were conducted using the KEGG database (http://www.genome.jp/kegg/) and Pathview software (32, 33).

## Measurement of membrane potential

Membrane potential was assessed using commercial kits following the protocol described (34). Briefly, bacterial cells were resuspended in M9 medium to an optical density at 600 nm ($OD_{600}$) of 0.2 and incubated at 37°C for 4 h with or without the addition of metabolites. For staining, 198 µL of the bacterial suspension was mixed with 2 µL of 10 mM $DiBAC_4(3)$ and incubated in the dark for 1 h. Finally, fluorescence was measured using a fluorescence plate reader (Biotek, Synergy HT, Vermont, USA) with excitation and emission wavelengths of 490 nm and 516 nm, respectively.

## Assessment of redox state

ROS levels were measured following the protocol described (35). Bacterial cells were resuspended in M9 medium to $OD_{600}$ of 0.2 and incubated at 37°C for 4 h with or without the addition of metabolites. After incubation, cells were collected, washed with PBS, and divided into two portions. One portion was analyzed for ROS. The samples were incubated with 20 µM DCFH-DA (Sigma, United States) at 37°C for 1 h in the dark. Fluorescence was measured using a microplate reader (CLARIO Star Plus, Germany) with excitation and emission wavelengths of 485 and 515 nm, respectively. The other portion was analyzed for total antioxidant capacity (T-AOC) using a ferric-reducing antioxidant power assay kit (Solarbio Science & Technology, Beijing, China). Absorbance was measured at 593 nm using a spectrophotometer.

## Measurement of enzyme activity

Bacteria were grown in M9 minimal medium supplemented with or without test metabolites for 5 h at 37°C. Cells were harvested by centrifugation (5,000 × $g$, 5 min, 4°C) and subjected to ultrasonic lysis with ice-bath cooling. The resulting lysates were clarified by centrifugation (12,000 × $g$, 10 min, 4°C) to remove cellular debris. Enzyme activities, including pyruvate dehydrogenase (PDH), α-ketoglutarate dehydrogenase (KGDH), succinate dehydrogenase (SDH), NADH, and NADPH were quantified using standardized commercial assay kits (Beyotime Biotechnology, China) according to manufacturer protocols. Enzyme activities (PDH, KGDH, SDH at 566 nm; NADH and NADPH at 450 nm) were measured spectrophotometrically using a microplate reader (Tecan Infinite E Plex). All experimental values were normalized to the untreated control group. Experiments were repeated in at least three independent biological replicates.

## Chloramphenicol quantification

Bacterial cultures were grown in the presence or absence of target metabolites using the method described above. Following harvest and ultrasonic lysis, the samples were subjected to chromatographic separation using an ExionLC liquid chromatography system (AB Sciex) coupled to a QTRAP 4500 mass spectrometer (AB Sciex). Chromatographic separation was performed on an Eclipse Plus C18 column (2.1 × 150 mm, 3.5 µm) maintained at 40°C. An injection volume of 10 µL was used with a mobile phase consisting of (A) water and (B) methanol, delivered at a flow rate of 0.30 mL/min. The gradient elution program was as follows: 0–5 min, 90% A to 5% A (10% B to 95% B); 5–7 min, 5% A (95% B); 7–9 min, 5% A to 90% A (95% B to 10% B). Mass spectrometric detection was conducted using an electrospray ionization source in negative ion mode with the following parameters: ion spray voltage, −4,500 V; source temperature, 550°C; collision gas, nitrogen; curtain gas, nitrogen at 241 kPa. Relative chloramphenicol levels were determined based on the corresponding peak areas.

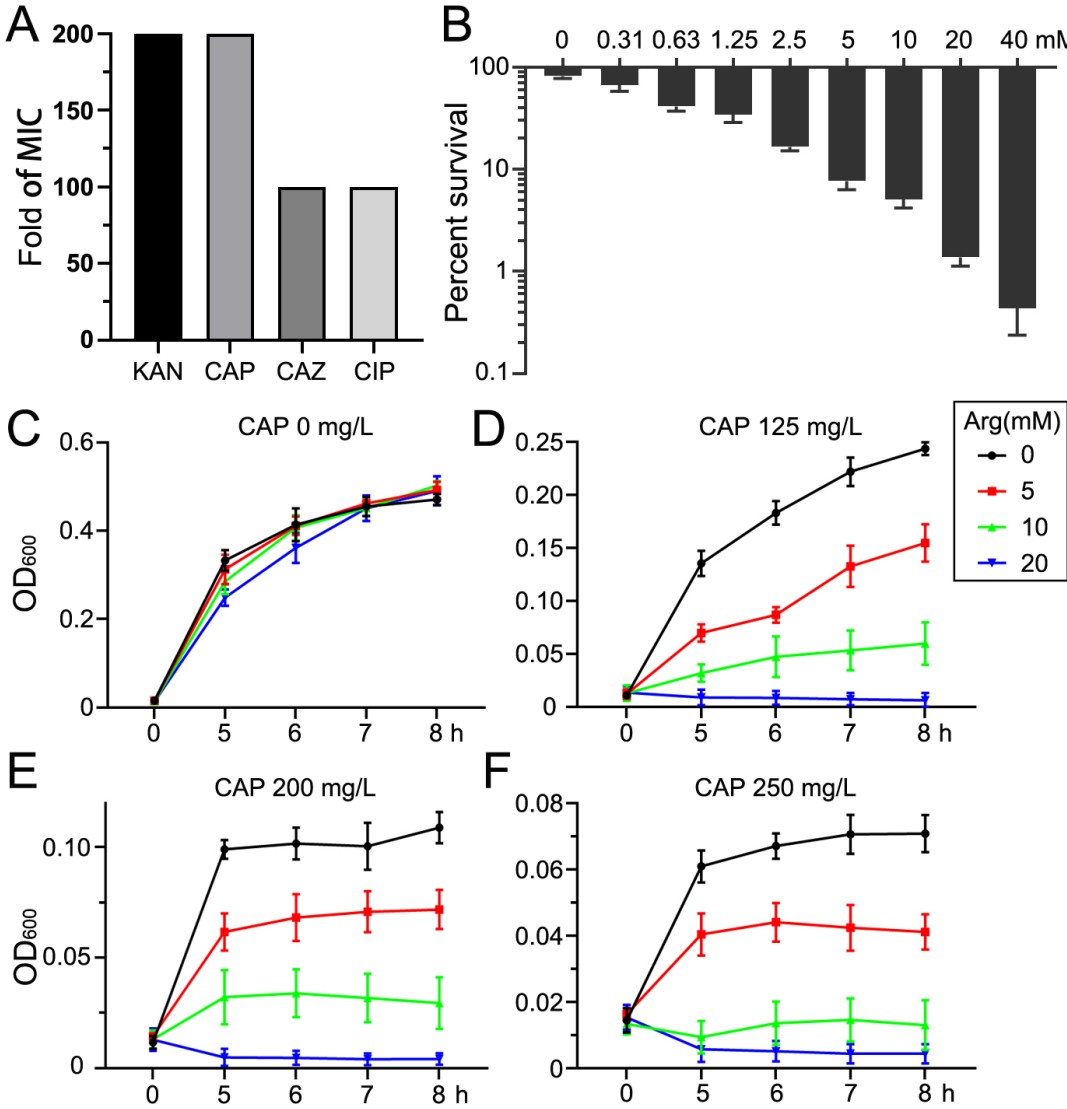

FIG 1 L-arginine enhances the bactericidal activity of chloramphenicol against multidrug-resistant *E. tarda*. (A) Fold increase in the MIC of multidrug-resistant *E. tarda* compared to the sensitive strain. (B) Effect of varying concentrations of Arg on bacterial survival. The *y*-axis represents the survival percentage of *E. tarda* with or without Arg under chloramphenicol pressure, while the *x*-axis indicates the concentration of Arg in M9 medium. The concentration of CAP is 0.8 mg/mL. (C–F) Impact of different concentrations of Arg and CAP on the growth of *E. tarda* in TSB medium. KAN, Kanamycin; CAP, Chloramphenicol; CAZ, Ceftazidime; CIP, Ciprofloxacin.

## RESULTS

### Arg enhances the bactericidal activity of CAP against multidrug-resistant *E. tarda*

The *E. tarda* strain used in this study exhibited multidrug resistance to kanamycin (KAN), CAP, ceftazidime (CAZ), and ciprofloxacin (CIP) (Fig. 1A). The survival rate of *E. tarda* decreased progressively with increasing concentrations of Arg. Under CAP treatment, survival dropped from 81.9% in the absence of Arg to 0.437% with 40 mM Arg, representing a 187-fold reduction compared to the control (Fig. 1B). The line chart further highlights the enhanced bacterial growth inhibition achieved with Arg under varying CAP concentrations (Fig. 1C through F). Notably, Arg alone did not exhibit any inhibitory effect on bacterial growth in TSB medium. However, the combination of Arg and CAP, particularly at an Arg concentration of 20 mM, completely suppressed bacterial growth.

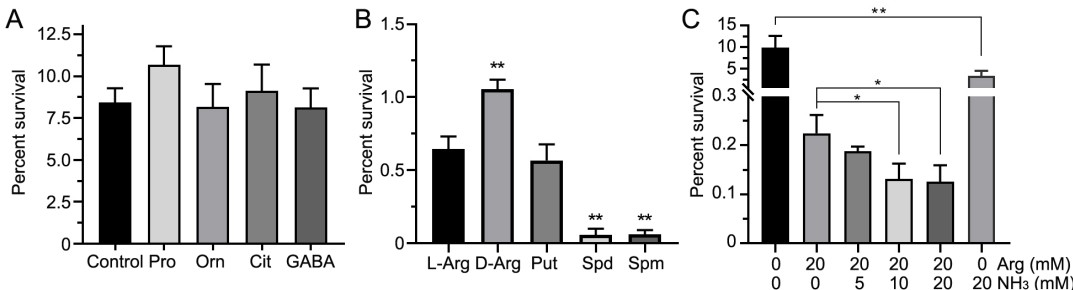

**FIG 2** Metabolites related to the arginine pathway enhance sensitivity to chloramphenicol. (A–C) The *y*-axis represents the survival percentage of *E. tarda* with or without metabolites under chloramphenicol pressure. The control group contains chloramphenicol without added metabolites, while experimental groups include chloramphenicol (1.0 mg/mL) along with the indicated metabolites (10 mM). Metabolites (Pro, Orn, Cit, GABA, L-Arg, D-Arg, Put, Spd, Spm, and NH₃) are shown on the *x*-axis. Statistical significance is indicated (*/**, *P* < 0.05/0.01 compared with control). Pro, proline; Orn, ornithine; Cit, citrulline; GABA, gamma-aminobutyric acid; L-Arg, L-arginine; D-Arg, D-arginine; Put, putrescine, Spd, spermidine; Spm, spermine.

## Metabolites associated with the Arg pathway increase sensitivity to CAP

The Arg-related metabolic pathway encompasses both Arg biosynthesis and catabolism. Among the tested metabolites, proline, ornithine, citrulline, and gamma-aminobutyric acid showed no significant impact on the sensitivity of *E. tarda* to CAP (Fig. 2A). In contrast, L-arginine, D-arginine, putrescine, spermidine, and spermine enhanced CAP's bactericidal activity (Fig. 2B). Notably, spermidine, and spermine exhibited the most pronounced effects, increasing bactericidal efficiency by 11.3-fold and 17.7-fold, respectively, compared to Arg. Furthermore, ammonia improved sensitivity to CAP, reducing the survival rate from 9.90% to 3.32% at 20 mM (2.98-fold decrease) (Fig. 2C). However, this enhancement was less significant than that observed with Arg, which reduced survival to 0.224% at 20 mM (44.1-fold decrease). The potentiation effect of Arg on CAP activity exhibited partial dependence on environmental pH. Alkaline conditions generated by NH₃ (20 mM) showed limited but statistically significant (*P* < 0.05) potentiation of Arg's antibiotic activity, further reducing survival from 0.224% (20 mM Arg) to 0.126% (20 mM Arg + 20 mM NH₃), representing a 1.79-fold increase in potency (Fig. 2C). Consequently, this pH-mediated modulation represented a secondary effect compared to Arg's primary potentiation, as evidenced by the 44-fold efficacy enhancement with 20 mM Arg alone compared to the 2.98-fold reduction achieved by 20 mM NH₃ alone.

## Arginine inhibits the TCA cycle

Given that TCA cycle activation has been demonstrated to enhance CAP resistance in *E. tarda* (20), we investigated whether Arg alters antibiotic efficacy via modulation of this metabolic pathway. Targeted LC–MS/MS analysis was performed using a triple quadrupole mass spectrometer to quantify changes in TCA cycle metabolite levels upon exogenous Arg supplementation. Untargeted profiling revealed marked increases in six key TCA cycle metabolites: citrate, α-ketoglutarate, succinate, fumarate, malate, and oxaloacetate (Fig. 3A). Enzymatic activity assays revealed that Arg suppressed the activities of PDH, KGDH, and SDH (Fig. 3B). α-Ketoglutarate (αKg), key metabolites in the TCA cycle, was found to increase the resistance of *E. tarda* to CAP, the percent survival from 0.834% to 94.0% (Fig. 3C). Even a small amount of αKg was sufficient to counteract the effect of Arg in enhancing the bactericidal efficiency of antibiotics. The presence of equal concentrations (20 mM) of αKg with Arg conferred CAP resistance in *E. tarda*. (Fig. 3D). Compared to the wild-type BW25113 strain, deletion mutants of isocitrate dehydrogenase (Δ*icd*) and α-ketoglutarate dehydrogenase (Δ*sucA*) exhibited heightened susceptibility to CAP, with OD₆₀₀ values decreasing from 0.609 (wild-type) to 0.435 (Δ*icd*) and 0.306 (Δ*sucA*) after 6 h of treatment (Fig. 3E). Notably, Arg supplementation potentiated CAP's inhibitory effects in all three strains.

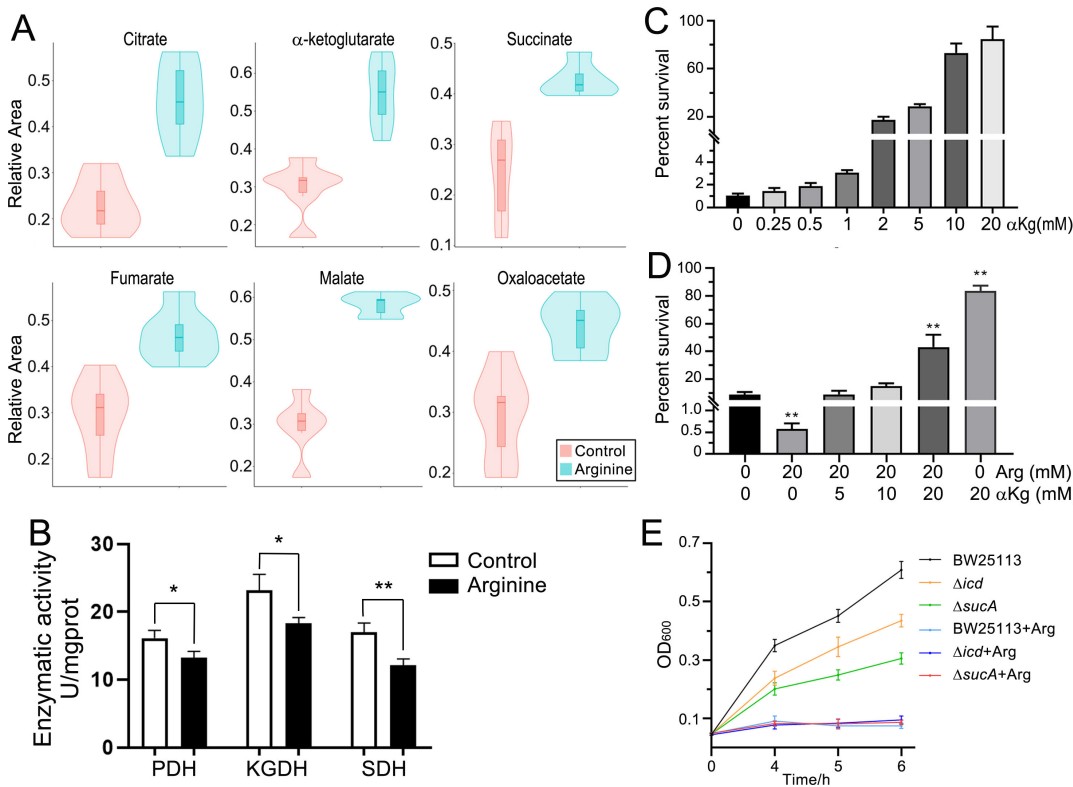

**FIG 3** The mutual influence between the TCA Cycle and Arg. (A) Exogenous addition of arginine leads to an increase in the content of metabolites in the TCA cycle. The *y*-axis represents the relative abundance of metabolites compared to the control group. (B) Enzymatic activity in the TCA cycle is suppressed by arginine. (C) α-Ketoglutarate reduces the sensitivity of bacteria to chloramphenicol (1.2 mg/mL). (D) α-Ketoglutarate reverses the bactericidal effect of L-arginine. (E) The influence of arginine on the antibiotic sensitivity of gene-deleted strains. The concentration of CAP in LB media is 0.2 mg/mL. (*/**, *P* < 0.05/0.01 compared with control) αKg, α-ketoglutarate; Arg, L-arginine. PDH, pyruvate dehydrogenase; KGDH, α-ketoglutarate dehydrogenase; SDH, succinate dehydrogenase.

## Metabolic profiling following exogenous Arg treatment

The suppressed TCA cycle alone does not fully account for the bactericidal enhancement by Arg. We, therefore, employed metabolomics to investigate its impact on other metabolic pathways. The metabolites in the exogenous Arg group and the control group were analyzed using LC-MS/MS. Principal component analysis revealed clear separation between the two groups (Fig. 4A). A total of 787 metabolites were identified, among which 291 metabolites exhibited significant differences in the L-Arg group compared to the control group. Of these, 211 were upregulated, while 80 were downregulated (Fig. 4B). The heatmap showed distinct abundance patterns of metabolites between the two groups (Fig. 4C). KEGG enrichment analysis identified three altered pathways (*P* < 0.05): phenylalanine metabolism, degradation of aromatic compounds, and the TCA cycle (Fig. 4D). In the TCA cycle, three metabolites showed significant changes: acetyl-CoA and citrate were upregulated, while thiamin pyrophosphate was downregulated. Within the phenylalanine metabolism pathway, eight metabolites were upregulated, including phenylpyruvate, trans-cinnamate, phenylglyoxylate, phenylacetylglycine, trans-3-hydroxycinnamate, trans-2-hydroxycinnamate, *N*-acetyl-L-phenylalanine, and acetyl-CoA (Fig. 4E).

## Modulation of CAP antibacterial efficacy and cellular redox status by Arg and Cin

trans-Cinnamate (Cin), identified as a differential metabolite, serves as a critical intermediate in phenylalanine metabolism. Our biochemical analyses revealed opposing

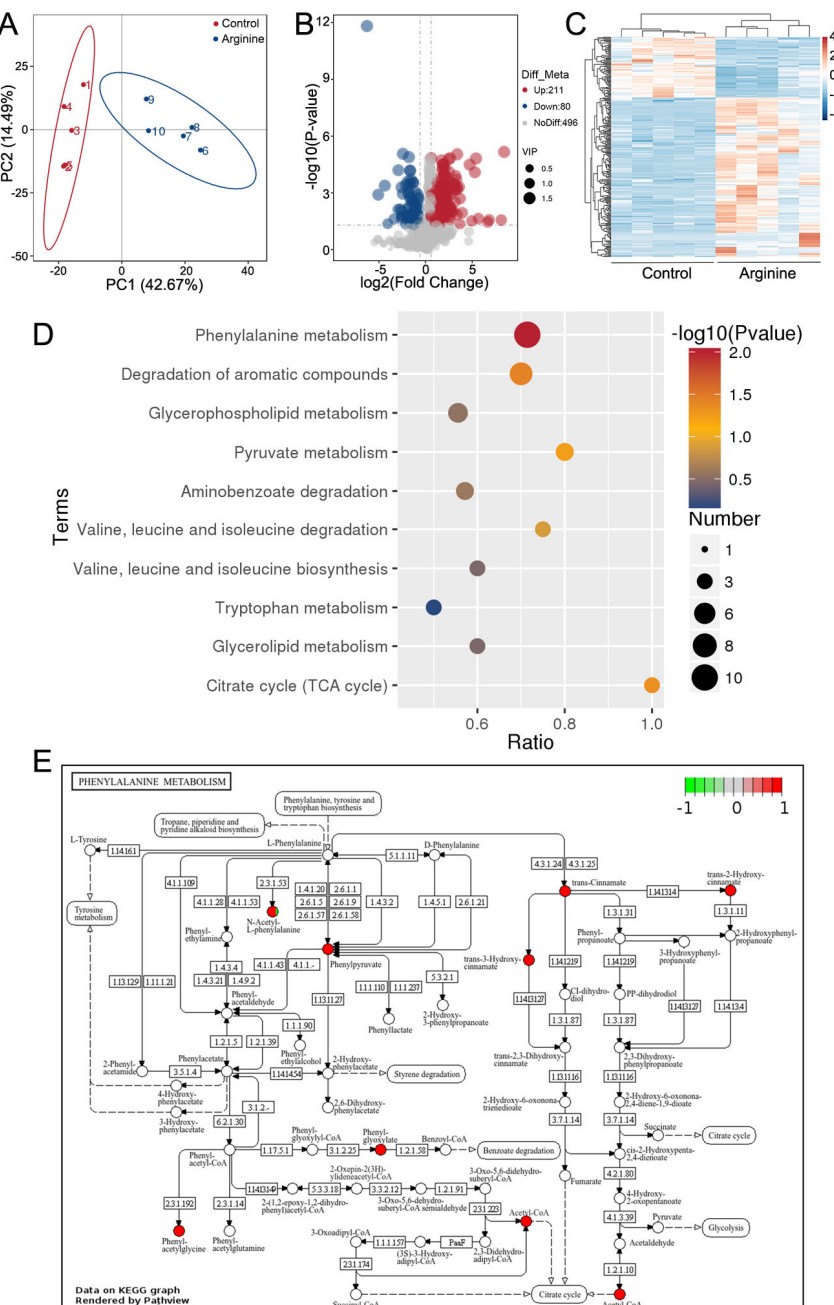

FIG 4 Metabolomic analysis of differential metabolites. (A) Principal component analysis of the control group and the exogenous L-arginine group. Each dot represents a biological replicate. (B) Volcano plot illustrating the differential metabolites between the control and exogenous arginine groups. The x-axis represents the fold change of metabolites, while the y-axis indicates statistical significance. Blue, red, and gray dots represent downregulated, upregulated, and non-significant metabolites, respectively. (C) Heatmap of altered metabolites in the presence or absence of L-arginine. Blue and red colors denote low and high metabolite abundance, respectively. (D) KEGG enrichment pathways. The P-value (−log10) of each pathway is represented by color, ranging from red (high significance) to blue (low significance). The diameter of the circles corresponds to the number of altered metabolites in each pathway. The x-axis indicates the ratio of altered metabolites in each pathway. (E) Phenylalanine metabolism pathway significantly enriched in the exogenous Arg group compared with the control group. Red dots represent metabolites with increased abundance.

effects between Arg and Cin on bacterial physiology. PMF measurements demonstrated that Arg-mediated depolarization of membrane potential facilitated intracellular CAP accumulation (Fig. 5A and B). Exogenous Arg supplementation attenuated cellular antioxidant capacity, as evidenced by total antioxidant capacity (T-AOC) (Fig. 5C), along with elevated ROS accumulation (Fig. 5D) and decreased NADH/NADPH levels (Fig. 5E and F). In striking contrast, Cin exhibited diametrically opposite effects: it enhanced antioxidant enzyme activities while maintaining membrane potential, consequently limiting antibiotic uptake.

## Arginine-dependent potentiation of chloramphenicol efficacy is modulated by Cin, oxidizing agents, and reducing agents

Consistent with α-ketoglutarate, Cin was observed to enhance *E. tarda* resistance to CAP (Fig. 6A). At equimolar concentrations (20 mM Arg + 20 mM Cin), bacterial survival increased, indicating an attenuation of antibiotic sensitivity (Fig. 6B). At concentrations of 5 mM, NADH failed to reverse the antibiotic-potentiating effect of Arg (Fig. 6C). While NADH treatment alone increased bacterial survival from 5.45% to 28.6%, this protective effect was markedly weaker than that observed with either NADPH or GSH alone under equivalent conditions. Simultaneously, the combination of 10 mM NADPH with 10 mM Arg resulted in significantly higher survival rates compared to the control (Fig. 6D). Among the tested redox-related metabolites (NADH, NADPH, and GSH), GSH exhibited the most potent capacity to counteract Arg-mediated potentiation of CAP activity. At 5 mM, GSH completely abolished the bactericidal enhancement induced by 20 mM Arg (Fig. 6E). Conversely, $H_2O_2$ dramatically enhanced CAP bactericidal efficacy (Fig. 6F). This

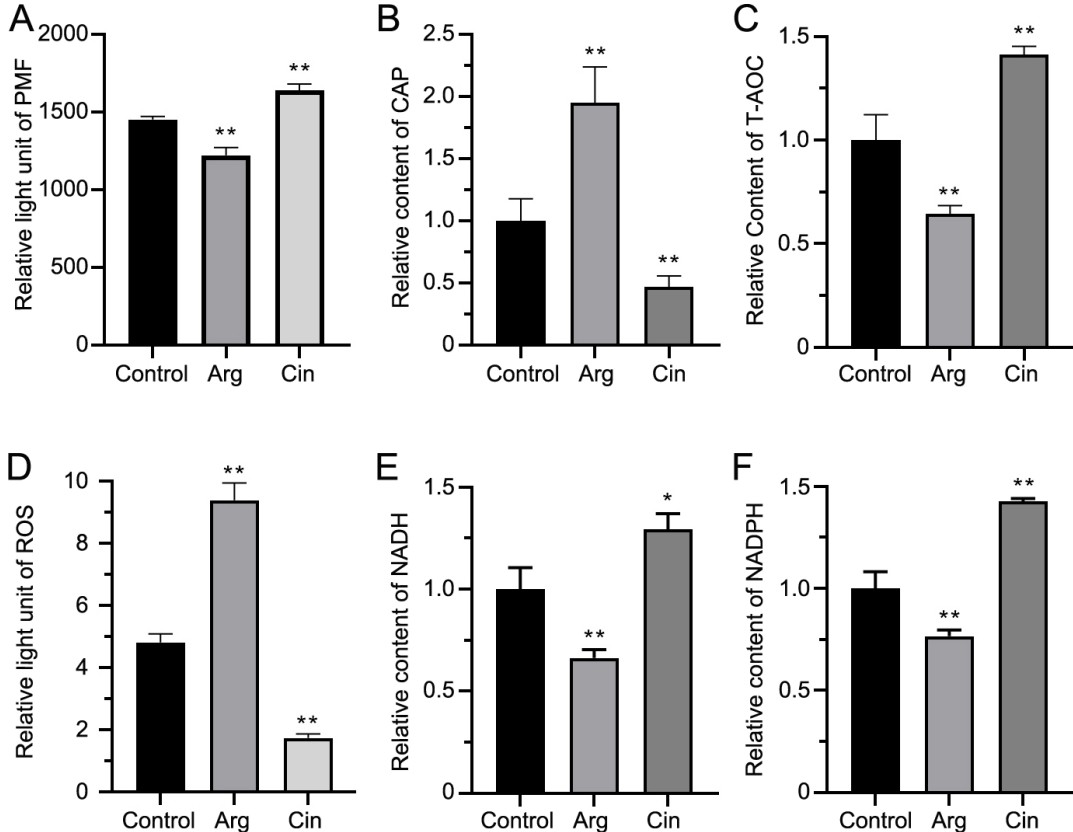

FIG 5 The influence of Arg and Cin on the PMF and redox state. Effects of L-arginine and trans-cinnamate on PMF (A), intracellular CAP (B), T-AOC (C), ROS (D), NADH (E), and NADPH (F). The *y*-axis represents the relative enzyme activity or content of PMF and intracellular chloramphenicol, under the influence of L-arginine (10 mM) and trans-cinnamate (10 mM) compared to the control group. (*/**, *P* < 0.05/0.01 compared with control) Arg, L-arginine; Cin, trans-cinnamate; PMF, proton motive force; CAP, chloramphenicol; T-AOC, total antioxidant capacity; ROS, reactive oxygen species.

pro-oxidant effect synergized with Arg, resulting in further potentiation of antibiotic activity beyond either treatment alone.

## DISCUSSION

Research on Arg as an antibiotic adjuvant has primarily centered on aminoglycosides, with studies revealing that its ability to enhance bactericidal activity against pathogens such as *P. aeruginosa* and *S. aureus* is modulated by pH and proton PMF (29). Additionally, as a precursor for NO synthesis, Arg was reported to increase *V. alginolyticus* resistance to gentamicin through NO-mediated mechanisms (30). Arg also improved the bactericidal efficiency of ciprofloxacin and tobramycin against *P. aeruginosa* although the underlying mechanisms remain unclear (36).

In this study, we observed that Arg enhanced the bactericidal efficacy of CAP against multidrug-resistant *E. tarda* in both M9 minimal and TSB medium. In the pathway of Arg and proline metabolism, certain metabolites such as proline, ornithine, citrulline, and gamma-aminobutyric acid did not alter CAP's bactericidal efficiency. In contrast, L-Arg, D-Arg, putrescine, spermidine, and spermine notably increased CAP's bactericidal activity, with spermidine and spermine exhibiting even greater enhancement than Arg. D-Arg acts as an inhibitor of Arg decarboxylase, blocking the decarboxylation of L-Arg and thereby preventing the synthesis of downstream polyamines, including putrescine, spermidine, and spermine, in the Arg metabolic pathway. The similar bacterial survival rates observed with the exogenous addition of L-Arg or D-Arg suggested that L-Arg did not enhance the bactericidal efficacy of CAP through polyamine synthesis. Furthermore, previous studies have demonstrated that polyamines primarily influence the bactericidal activity of antibiotics through two mechanisms. First, polyamines mitigate ROS-induced damage by directly scavenging ROS, enhancing the activity of antioxidant enzymes such as superoxide dismutase and glutathione peroxidase, promoting the expression of antioxidant genes via pathways such as Nrf2, and producing antioxidants like NADPH and glutathione. These actions collectively enhance cellular antioxidant capacity (37, 38). Second, polyamines contribute to cell membrane homeostasis and protection against oxidative stress (39, 40). For instance, polyamines have been reported to enhance *E. coli* resistance to fluoroquinolones, aminoglycosides, and cephalosporins by scavenging ROS (41). In contrast to these findings, this study revealed that polyamines increased bacterial sensitivity to CAP. This discrepancy highlighted a distinct mechanism and supports the conclusion that L-Arg did not modulate the bactericidal activity of CAP through polyamine metabolism.

Given previous findings that pH influences antibiotic efficacy (29) and that many Arg metabolic intermediates (e.g., putrescine, spermidine, and spermine) are polyamines with alkaline properties, we investigated whether pH might play a role in Arg's ability to enhance CAP's bactericidal activity. Ammonia, a basic compound, was observed to enhance the bactericidal effect of CAP, reducing bacterial survival by 2.98-fold. However, this effect was markedly weaker than the 44.1-fold reduction mediated by Arg. Similarly, ornithine, another basic amino acid with a pKa of ~9.71 (slightly lower than Arg's pKa of ~10.76), did not enhance CAP's bactericidal efficiency. These findings indicate that while pH changes induced by basic metabolites like ammonia may contribute to CAP's bactericidal activity to a certain extent, the effect is relatively limited and cannot fully account for Arg's potent enhancement. This suggests that pH modulation is not the primary mechanism by which Arg promotes CAP's bactericidal efficacy.

Non-targeted metabolomics analysis revealed significant perturbations in the TCA cycle following Arg supplementation, particularly showing marked upregulation of entry-point metabolites including acetyl-CoA and citrate. Subsequent targeted metabolite quantification confirmed the coordinated elevation of six TCA cycle intermediates. Intriguingly, we observed downregulation of thiamine pyrophosphate, an essential cofactor for both pyruvate dehydrogenase and αKg dehydrogenase complexes, suggesting functional impairment of the TCA cycle. This metabolic reprogramming appears mechanistically linked to enhanced antibiotic susceptibility, as evidenced by two key

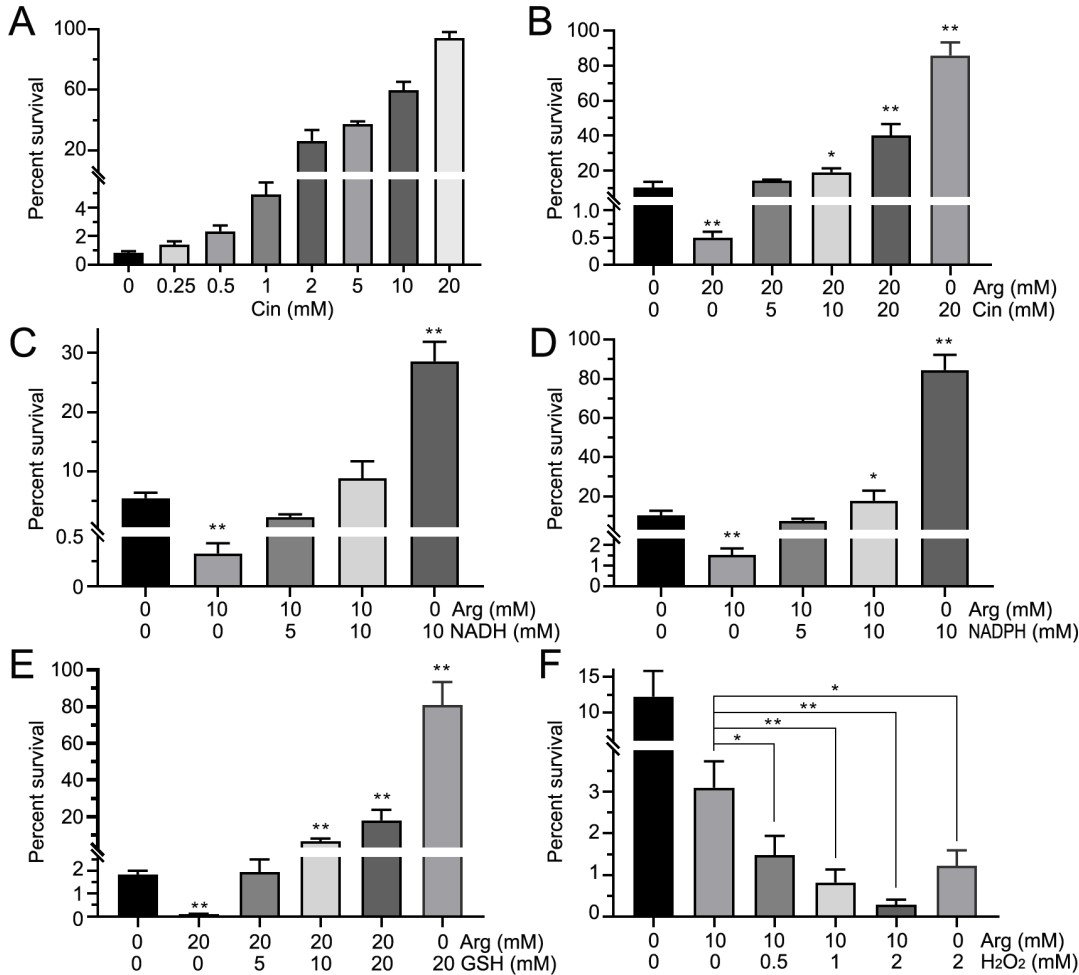

**FIG 6** The impact of cinnamate, oxidizing agents, and reducing agents on arginine-potentiated chloramphenicol bactericidal activity percent survival of bacteria under different concentrations of Cin and Arg (A and B). Metabolites, including NADH (C), NADPH (D), GSH (E), and $H_2O_2$ (F), affect the bactericidal effect of L-arginine. The $y$-axis represents the percent survival of bacteria with or without metabolites under chloramphenicol pressure. The $x$-axis indicates the concentrations of trans-cinnamate, L-arginine, NADH, NADPH, GSH, and $H_2O_2$ (mM). The concentration of CAP is 1.0 mg/mL. (*/**, $P < 0.05/0.01$ compared with control). Cin, trans-cinnamate; Arg, L-arginine; GSH, glutathione.

findings: (i) αKg was shown to promote CAP resistance while abolishing Arg's bactericidal enhancement; (ii) the Δicd and ΔsucA deletion mutant (defective in isocitrate dehydrogenase or αKg dehydrogenase) exhibited increased sensitivity to CAP. Collectively, these results suggest that the TCA cycle inhibition by Arg underlies its enhancement of CAP's bactericidal effect.

Arg was found to similarly enhance CAP sensitivity in both deletion strains (Δicd and ΔsucA) and BW25113, indicating that TCA cycle inhibition is not the sole mechanism underlying Arg's antibiotic-potentiating effects. Furthermore, untargeted metabolomics revealed that exogenous Arg supplementation reprogrammed phenylalanine metabolism, resulting in the upregulation of all eight detected metabolites. The influence of phenylalanine metabolism on the bactericidal efficacy of antibiotics remains underexplored. Previous studies have shown that phenylalanine exerts varying effects on different antibiotics (42), and Cin has been reported to enhance the bactericidal activity of β-lactam antibiotics, although the underlying mechanisms remain unclear (43). Cin, a key metabolite in the phenylalanine metabolism pathway, demonstrated a concentration-dependent effect on bacterial resistance to CAP, with increasing concentrations of Cin leading to a gradual rise in resistance at lower concentration ranges. Additionally, the ability of Arg to enhance the bactericidal efficiency of CAP was markedly inhibited by

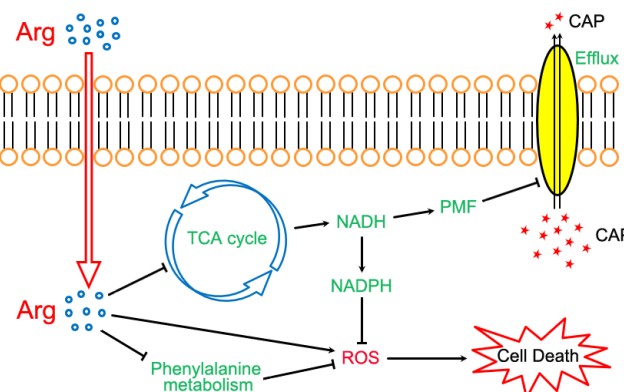

**FIG 7** Proposed mechanism of L-arginine enhancing the bactericidal efficiency of chloramphenicol against *E. tarda*. Red indicates increased metabolite content, while green represents decreased metabolite content or inhibited pathways. Arg, L-arginine; TCA, tricarboxylic acid; NADH, nicotinamide adenine dinucleotide; NADPH, nicotinamide adenine dinucleotide phosphate; PMF, proton motive force; CAP, chloramphenicol; PhM, phenylalanine metabolism; ROS, reactive oxygen species.

Cin. Moreover, we discovered that Cin not only reduces antibiotic uptake into bacterial cells by enhancing the PMF but also mitigates antibiotic-induced oxidative damage by boosting the bacterial antioxidant capacity. PMF provides the energy required for drug efflux systems, which are essential for CAP resistance (44, 45). In a contrasting mechanism to Arg, exogenous Cin increases the PMF in *E. tarda*, thereby enhancing the energy supply for efflux systems and reducing intracellular antibiotic accumulation. Interestingly, while aminoglycosides rely on PMF to facilitate antibiotic uptake (28), this contrast underscores the unique mechanism by which Arg enhances *E. tarda* sensitivity to CAP. Collectively, these findings suggested that Arg enhanced the bactericidal efficacy of CAP by interfering with phenylalanine metabolism, particularly through the inhibition of Cin metabolism.

TCA cycle inhibition reduced NADH and ATP production—critical components for maintaining cellular reductants including NADPH and glutathione (42). Glutathione serves as a critical component of the bacterial antioxidant defense system (27, 46, 47). In multidrug-resistant *E. tarda*, pyruvate reprograms glycine, serine, and threonine metabolism, resulting in coordinated upregulation of glutathione peroxidase activity and concurrent suppression of glutathione reductase. This shift in redox enzyme activity disrupts cellular redox homeostasis, leading to accumulation of intracellular reactive oxygen species (ROS) and ultimately restoring antibiotic susceptibility (28). In our study, both Arg and Cin modulate redox homeostasis by altering levels of ROS, NADH, and NADPH. Furthermore, bactericidal assays with exogenous $H_2O_2$ and reducing agents confirmed that oxidizing agents enhance Arg-potentiated chloramphenicol efficacy, whereas reducing agents abolish it. These results demonstrate that Arg influences CAP bactericidal efficiency primarily through redox regulation.

Based on our findings, Arg was shown to enhance the sensitivity of *E. tarda* to CAP through three distinct mechanisms (Fig. 7). First, Arg inhibited the TCA cycle, reducing the production of NADH and PMF. This led to insufficient energy for antibiotic efflux and a deficiency in hydrogen donors required for reductants such as NADPH and glutathione. Second, Arg increased ROS levels while decreasing T-AOC, amplifying oxidative damage caused by CAP. Third, Arg further potentiated the effects of these two metabolic pathways by disrupting phenylalanine metabolism. Elevated levels of cellular reductants (NADH and NADPH) combined with enhanced PMF are known to decrease CAP susceptibility. However, L-arginine effectively counteracts this resistance phenotype through the suppression of the TCA cycle and phenylalanine metabolism.

Building upon the concept of metabolome reprogramming (48), our study demonstrated that Arg sensitized multidrug-resistant *E. tarda* to CAP by modulating redox

balance and inhibiting the TCA cycle and phenylalanine metabolism. These findings broaden the understanding of how metabolic modulation influences the bactericidal efficacy of antibiotics and provide a solid theoretical foundation for developing therapeutic strategies to combat antibiotic-resistant bacteria.

## ACKNOWLEDGMENTS

This work was financially supported by Natural Science Foundation of Shandong Province (ZR2022MC170, ZR2019PH019, and ZR2022MH235), Special Funding for High-Level Talents in the Medical and Health Industry of Jinan (202412), Key Research and Development Program of Shandong Province (2024LZGC024), and Modern Agricultural Fish Industry Technology System of Shandong Province (SDAIT-12-04, SDAIT-12-09).

Conceptualization: C.W., X.-Y.L., and B.-B.Y. Methodology: B.-B.Y., N.L., Y.Z., X.-S.D., X.X., and Q.-L.M. Investigation: B.-B.Y., N.L., Y.Z., X.-S.D., X.X., L.A., X.-R.W., L.Y., and Q.-L.M. Software: N.L., X.-S.D., L.-L.K., and X.X. Funding acquisition: C.W., B.-B.Y., and X.-Y.L. Project administration: C.W. Writing—original draft: C.W. and B.-B.Y. Writing—review and editing: C.W., B.-B.Y., and X.-Y.L.

## AUTHOR AFFILIATIONS

[1]Department of Neonatology, Children's Hospital Affiliated to Shandong University, Jinan, China
[2]Department of Neonatology, Jinan Children's Hospital, Jinan, China
[3]Department of Genetics and Breeding, Shandong Freshwater Fisheries Research Institute, Jinan, China
[4]Pharmacy Department, The First People's Hospital of Jinan, Jinan, China

## AUTHOR ORCIDs

Xiao-ying Li http://orcid.org/0009-0001-2013-7624
Chao Wang http://orcid.org/0000-0002-1113-259X

## AUTHOR CONTRIBUTIONS

Bei-bei Yan, Formal analysis, Investigation, Writing – original draft, Writing – review and editing | Na Li, Investigation | Yang Zhou, Formal analysis, Investigation | Li-li Kang, Investigation | Xue-sa Dong, Formal analysis, Investigation | Li An, Investigation | Qing-lei Meng, Investigation | Xi-rong Wang, Investigation | Ling Yang, Investigation | Xiao-ying Li, Funding acquisition, Writing – review and editing.

## DATA AVAILABILITY

The raw metabolomics data generated in this study have been deposited in the Metabolomics Workbench under Project ID ST003866.

## ADDITIONAL FILES

The following material is available online.

### Open Peer Review

PEER REVIEW HISTORY (review-history.pdf). An accounting of the reviewer comments and feedback.

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
