## [Reviewer comments · mSystems]

Metabolic Potentiation of Antibiotic Killing by L-Arginine in Drug-Resistant *Edwardsiella tarda*

Bei-bei Yan, Na Li, Yang Zhou, Li-li Kang, Xuesu Dong, Xiao Xu, Li An, Qinglei Meng, Xi-rong Wang, Ling Yang, Xiaoying Li, and Chao Wang

Corresponding Author(s): Chao Wang, Shandong Freshwater Fisheries Research Institute

Review Timeline:

Submission Date:	October 21, 2025
Editorial Decision:	November 14, 2025
Revision Received:	November 24, 2025
Accepted:	December 7, 2025

Editor: Lennart Schada von Borzyskowski

Reviewer(s): The reviewers have opted to remain anonymous.

Transaction Report:

DOI: <https://doi.org/10.1128/msystems.01509-25>

Re: mSystems01509-25 (Metabolic Potentiation of Antibiotic Killing by L-Arginine in Drug-Resistant *Edwardsiella tarda*)

Dear Dr. Chao Wang:

I am happy to say that your manuscript is generally of interest to the readers of mSystems and might be accepted, provided that you can modify the manuscript in accordance with the provided reviewer comments.

Revision Guidelines

Sincerely,
Lennart Schada von Borzyskowski
Editor
mSystems

Reviewer #1 (Comments for the Author):

This manuscript presents a compelling study on the potentiation of chloramphenicol activity by L-arginine against multidrug-resistant *Edwardsiella tarda*. The authors demonstrate that L-arginine modulates the TCA cycle, redox homeostasis, and phenylalanine metabolism, ultimately enhancing chloramphenicol bactericidal efficacy by at least two orders of magnitude.

These findings significantly advance our understanding of metabolic regulation in antibiotic resistance and hold considerable relevance for the development of novel therapeutic strategies. While the manuscript is well-structured and the conclusions are generally supported, the following issues should be addressed to further improve its clarity and rigor:

1. Line 65: The species name "*Escherichia coli*" should be italicized.
2. Lines 163 and 172: The term "T-AOC" is used repeatedly; please clarify the methodological details.
3. Line 335: The abbreviation for "thiamine pyrophosphate" is unnecessary, as the term appears only once.
4. Figure 2D is referenced in the text but does not appear in the provided figures. Based on the descriptions in Lines 202 and 208, it appears that "Figure 2D" should be "Figure 2C." Please verify and correct this inconsistency.
5. In Figures 1C-1F, the axis labels are too small to be easily legible; please increase the font size for improved readability.
6. In Figure 7, a red line appears beneath "PhM"; please revise the figure to remove this artifact.

Reviewer #2 (Comments for the Author):

This study demonstrates that L-arginine significantly enhances the bactericidal efficacy of chloramphenicol against multidrug-resistant *E. tarda*. Through a combination of targeted metabolite analysis and untargeted metabolomics, the authors further reveal that arginine modulates both the TCA cycle and phenylalanine metabolism. They provide compelling evidence that these two metabolic pathways influence bacterial redox homeostasis and proton motive force, ultimately altering chloramphenicol's antibacterial activity. The identification of phenylalanine metabolism as a pathway affecting antibiotic efficacy represents a notable and innovative aspect of this work. Several issues need to be addressed to enhance the quality of this manuscript.

Major comments

- The discussion of glutathione and ROS in Line 82 provides illustrative examples but lacks mechanistic depth. The authors should elaborate on the specific metabolic pathways and regulatory mechanisms through which these factors influence antibiotic efficacy.
- Although the term "reprogramming" is repeatedly used throughout the manuscript, the authors do not provide a clear discussion of the current research status regarding metabolic reprogramming in the context of antibiotic resistance. The authors are encouraged to explicitly elaborate-based on their own findings-how metabolic reprogramming is manifested in *E. tarda* under arginine treatment, and through which specific regulatory nodes it influences chloramphenicol susceptibility.
- The authors should discuss the crucial role of metabolic reprogramming in coping with antibiotic resistance with updated literature.

Minor comments

- The concentration of CAP and other metabolites need to be annotated in many Figures.
- There are some mistakes that need to be revised: Figure 2D needs to be revised into 2C (Line 202 and 208); The term "metabolite" should be used in its plural form (Line 443); capitalizing the first letter in line 46.
- Why does the knockout strain use *Escherichia coli* instead of *Edwardsiella tarda*?
- In Figure 7, "PhM" is used as an abbreviation for "phenylalanine metabolism." The full term should be spelled out for clarity.
- Results 3.3 prove that arginine inhibits the TCA cycle, but the metabolites in the TCA cycle are all upregulated. How can this be explained?
- Lines 284-287 contain repetitive statements. Revising these sentences to eliminate redundancy would enhance the clarity of the paragraph.
- In many instances, the fonts for *Edwardsiella tarda*, *Escherichia coli*, etc., do not have italics.
- There is a formatting issue with reference 31.
- The methodology for determining chloramphenicol concentration is not provided in the manuscript.
- The manuscript uses a lot of "significantly", which is not a scientific word. Please remove it.
- Data analysis should be included as an individual section in the Materials and Methods.

Dear Editors and Reviewers,

We sincerely thank you for your time and valuable comments on our manuscript. Your feedback has been immensely helpful in guiding us to improve the quality and clarity of our work. We have carefully considered all the suggestions and have revised the manuscript accordingly. All changes have been highlighted in red for your convenience.

Below, we provide our point-by-point responses to the reviewers' comments:

Reviewer #1

1. Line 65: The species name "Escherichia coli" should be italicized.

Answer:

We have corrected "*Escherichia coli*" into italicized in Line 66.

2. Lines 163 and 172: The term "T-AOC" is used repeatedly; please clarify the methodological details.

Answer:

We have corrected the error by removing the repetitive content in Lines 187 and 190, as suggested.

3. Line 335: The abbreviation for "thiamine pyrophosphate" is

unnecessary, as the term appears only once.

Answer:

We have removed the abbreviation of ThPP in Line 362.

4. Figure 2D is referenced in the text but does not appear in the provided figures. Based on the descriptions in Lines 202 and 208, it appears that "Figure 2D" should be "Figure 2C." Please verify and correct this inconsistency.

Answer:

We have corrected the reference to "Figure 2D" to "Figure 2C" in Lines 230 and 237.

5. In Figures 1C-1F, the axis labels are too small to be easily legible; please increase the font size for improved readability.

Answer:

We have addressed this issue by increasing the font sizes in Figure 1 to enhance readability.

6. In Figure 7, a red line appears beneath "PhM"; please revise the figure to remove this artifact.

Answer:

We have revised "PhM" in Figure 7 to the full term "Phenylalanine

metabolism".

Reviewer #2

Major comments

- The discussion of glutathione and ROS in Line 82 provides illustrative examples but lacks mechanistic depth. The authors should elaborate on the specific metabolic pathways and regulatory mechanisms through which these factors influence antibiotic efficacy.

Answer:

As suggested, we have added content concerning glutathione and ROS in Lines 86 and 395.

- Although the term "reprogramming" is repeatedly used throughout the manuscript, the authors do not provide a clear discussion of the current research status regarding metabolic reprogramming in the context of antibiotic resistance. The authors are encouraged to explicitly elaborate-based on their own findings-how metabolic reprogramming is manifested in *E. tarda* under arginine treatment, and through which specific regulatory nodes it influences chloramphenicol susceptibility.

Answer:

As suggested, we have incorporated discussions on metabolic reprogramming throughout the manuscript, with relevant content added in Lines 14, 63, 97, 363, and 373.

-The authors should discuss the crucial role of metabolic reprogramming in coping with antibiotic resistance with updated literature.

Answer:

As suggested, we have expanded the coverage of metabolic reprogramming and antibiotic resistance with updated literature in both the introduction and discussion sections (Lines 68-70, 86-101, 395-407).

Minor comments

- The concentration of CAP and other metabolites need to annotate in many Figures.

Answer:

CAP and metabolite concentrations are now indicated in the figures.

- There some mistakes need to be revised: Figure 2D needs to be devised into 2C (Line 202 and 208); The term "metabolite" should be used in its plural form (Line 443); capitalizing the first letter in line 46.

Answer:

We thank the reviewer for their careful reading. The following corrections have been made: the figure citation in Lines 230 and 237 has been updated to Figure 2C; "metabolite" has been pluralized in Figure 2; and "Introduction" has been capitalized in Line 46.

- Why does the knockout strain use *Escherichia coli* instead of *Edwardsiella tarda*?

Answer:

While gene knockout techniques were not available for *Edwardsiella tarda* in our experimental system, we employed an established comparative approach using *Escherichia coli* gene deletion mutants from the KEIO collection. Given that *E. coli* and *E. tarda* both belong to the Enterobacteriaceae family and share conserved core metabolism, validation in this model organism provides strong supportive evidence for our mechanistic findings.

- In Figure 7, "PhM" is used as an abbreviation for "phenylalanine metabolism." The full term should be spelled out for clarity.

Answer:

Corrected as suggested: "PhM" in Figure 7 has been changed to "Phenylalanine metabolism".

- Results 3.3 proves that arginine inhibits the TCA cycle, but the metabolites in the TCA cycle are all upregulated. How can this be explained?

Answer:

Exogenous arginine supplementation suppressed the activity of key TCA cycle enzymes, including pyruvate dehydrogenase, α -ketoglutarate dehydrogenase, and succinate dehydrogenase. Consistently, metabolomic analysis revealed a decrease in thiamine pyrophosphate—an essential cofactor for both pyruvate dehydrogenase and α -ketoglutarate dehydrogenase complexes. The inhibition of these enzymes led to metabolic bottlenecks, resulting in the accumulation of upstream metabolites.

- Lines 284-287 contain repetitive statements. Revising these sentences to eliminate redundancy would enhance the clarity of the paragraph.

Answer:

We have rewritten the sentences in Line 312 to improve clarity.

- In many instances, the fonts for *Edwardsiella tarda*, *Escherichia coli*, etc., do not have italics.

Answer:

We have corrected the Latin names for bacteria to italics throughout the manuscript.

- There is a formatting issue with reference 31.

Answer:

We apologize for the formatting errors that appeared in the originally submitted PDF. These have now been corrected in the revised manuscript.

- The methodology for determining chloramphenicol concentration is not provided in the manuscript.

Answer:

The method for chloramphenicol quantification has been added to Section 2.8 (Line 194).

-The manuscript uses a lot of "significantly", which is not a scientific word. Please remove it.

Answer:

As recommended, the redundant use of "significantly" has been removed from the full text.

-Data analysis should be included as an individual section in the

Materials and Methods.

Answer:

Corrected as suggested: A "Metabolomic Data Analysis" section has been added at Line 154.

Re: mSystems01509-25R1 (Metabolic Potentiation of Antibiotic Killing by L-Arginine in Drug-Resistant *Edwardsiella tarda*)

Dear Dr. Chao Wang:

Your manuscript has been accepted, and I am forwarding it to the ASM production staff for publication. Your paper will first be checked to make sure all elements meet the technical requirements. ASM staff will contact you if anything needs to be revised before copyediting and production can begin. Otherwise, you will be notified when your proofs are ready to be viewed.

Sincerely,
Lennart Schada von Borzyskowski
Editor
mSystems

Reviewer #1 (Comments for the Author):

The authors have answered all my questions.

Reviewer #2 (Comments for the Author):

The authors sufficiently addressed my concerns.